# Stage- and Subfield-Associated Hippocampal miRNA Expression Patterns after Pilocarpine-Induced Status Epilepticus

**DOI:** 10.3390/biomedicines10123012

**Published:** 2022-11-23

**Authors:** Yue Li, S Thameem Dheen, Fengru Tang, Yumin Luo, Ran Meng, Tay Sam Wah Samuel, Lan Zhang

**Affiliations:** 1Department of Anatomy, Yong Loo Lin School of Medicine, National University of Singapore, Singapore 117597, Singapore; 2Department of Pharmacy, Xuanwu Hospital, Capital Medical University, Beijing 100053, China; 3Department of Education, Xuanwu Hospital, Capital Medical University, Beijing 100053, China; 4Radiation Physiology Lab, Singapore Nuclear Research and Safety Initiative, National University of Singapore, Singapore 138602, Singapore; 5Department of Neurology, Xuanwu Hospital, Capital Medical University, Beijing 100053, China

**Keywords:** epilepsy, cornu ammonis, dentate gyrus, miRNA expression pattern

## Abstract

Objective: To investigate microRNA (miRNA) expression profiles before and after pilocarpine-induced status epilepticus (SE) in the cornu ammonis (CA) and dentated gyrus (DG) areas of the mouse hippocampus, and to predict the downstream proteins and related pathways based on bioinformatic analysis. Methods: An epileptic mouse model was established using a pilocarpine injection. Brain tissues from the CA and DG were collected separately for miRNA analysis. The miRNAs were extracted using a kit, and the expression profiles were generated using the SurePrint G3 Mouse miRNA microarray and validated. The intersecting genes of TargetScan and miRanda were selected to predict the target genes of each miRNA. For gene ontology (GO) studies, the parent-child-intersection (pci) method was used for enrichment analysis, and Benjamini-Hochberg was used for multiple test correction. The Kyoto Encyclopedia of Genes and Genomes (KEGG) was used to detect disease-related pathways among the large list of miRNA-targeted genes. All analyses mentioned above were performed at the time points of control, days 3, 14, and 60 post-SE. Results: Control versus days 3, 14, and 60 post-SE: in the CA area, a total of 131 miRNAs were differentially expressed; 53, 49, and 26 miRNAs were upregulated and 54, 10, and 22 were downregulated, respectively. In the DG area, a total of 171 miRNAs were differentially expressed; furthermore, 36, 32, and 28 miRNAs were upregulated and 78, 58, and 44 were downregulated, respectively. Of these, 92 changed in both the CA and DG, 39 only in the CA, and 79 only in the DG area. The differentially expressed miRNAs target 11–1630 genes. Most of these proteins have multiple functions in epileptogenesis. There were 15 common pathways related to altered miRNAs: nine different pathways in the CA and seven in the DG area. Conclusions: Stage- and subfield-associated hippocampal miRNA expression patterns are closely related to epileptogenesis, although the detailed mechanisms need to be explored in the future.

## 1. Introduction

Epilepsy is associated with plasticity of synaptic transmission, axonal sprouting, and neurogenesis in the hippocampus [1,2]. The hippocampus is divided into five sub-regions; CA1, CA2, CA3, CA4 (Cornu Ammonis region, Ammon’s horn), and the dentate gyrus (DG), based on cell size, shape, connectivity, and function [3]. Granule neurones are mainly distributed in the DG, whereas pyramidal neurones are predominantly concentrated in the CA region. These regions are distinguished by their unique patterns of connectivity; layer II neurones project to the DG and CA3/CA2 of the hippocampus proper, whereas layer III neurones project to CA1 and the subiculum [4].

Animal models have been used extensively to study the mechanisms of epilepsy; these include models of pilocarpine-induced status epilepticus (SE), electrical stimulation-induced SE, or kainic acid (KA)-induced SE. The ability of pilocarpine to induce SE is due to the activation of the M1 muscarinic receptor subtype. Following pilocarpine-induced SE, the mortality rate of adult male albino mice was 25–50%, the mean latent period was 14.4 days, and spontaneous recurrent seizures (SRSs) lasted 50–60 s with a frequency of 1–5 seizures per week in the chronic stage. Injection of pilocarpine induces an SE characterised by tonic-clonic generalised seizures, which spontaneously remit several hours later; this stage was defined as the acute stage with loss of neurones (days 1 to 3 post-SE), after which the mice went into a seizure-free period called the latent period with more prominent loss of neurones (days 7 to 14 post-SE), before displaying the SRSs and entering chronic epileptic condition characterised by an increase of astrocytes in the DG known as sclerosis (day 60 post-SE) [5,6]. The key cellular components of hippocampal formation, DG granule cells, and CA neurones are strikingly different during chronic epileptic activity. The DG has a uniquely permissible environment for neurogenesis, whereas the CA does not. Multipotent neuroprogenitor cells differentiate only into neurones when grafted into the DG area [7]. These characteristics prompted us to study these two parts separately.

MicroRNAs (miRNAs) are small RNAs that regulate gene expression by binding to the 3′-UTR of mRNAs, causing translational repression and/or mRNA degradation [8,9]. The mechanisms of this involve binding to target mRNAs and reducing protein production via Argonaute (AGO)-mediated translational repression [8]. Approximately 1500 miRNAs in the human genome are predicted to regulate more than one-third of all protein-coding genes because a single miRNA can target approximately 200 mRNAs [10].

Current studies on miRNA expression in epilepsy involve the analysis of blood, cerebrospinal fluid, and brain tissues in both animal models and patients, in terms of pathogenesis, diagnosis, treatment, and prognosis [11,12,13,14,15,16,17]. However, the miRNA expression profiles in the CA and DG areas are still relatively unknown. Herein, the expression profile of miRNAs in the CA and DG were examined based on miRNA microarrays and confirmed with quantitative real-time polymerase chain reaction (qRT-PCR) and bioinformatics analysis.

## 2. Materials and Methods

### 2.1. Sample Preparation

#### 2.1.1. Experimental Animals

This study was approved by the Institutional Animal Care and Use Committee of the National University of Singapore (IACUC approved protocol No: B27/09&098/09). Experiments were performed on 8–10-week-old non-spontaneously epileptic adult male Swiss albino mice weighing in a range from 25 g–30 g. The mice used in the study were randomly divided into two groups: the control group did not undergo pilocarpine injection but saline, whereas the pilocarpine-induced seizures group underwent pilocarpine injection. All mice in the control and pilocarpine-induced seizures group were sacrificed, and the latter group was re-divided into three subgroups according to the time points at which they were sacrificed; days 3 (acute), 14 (latency), and 60 (chronic) post-SE. Three mice were included in each group.

#### 2.1.2. Pilocarpine-Induced SE Mouse Model Preparation

The mice were administered a single subcutaneous injection of methyl-scopolamine nitrate (1 mg/kg) 30 min before the injection of either saline in the control or pilocarpine (300 mg/kg) in the experimental groups. Treated mice experienced acute SE. If the mice did not experience continuous seizures for 4 h, they were considered non-SE mice and were excluded from our study. Some mice experienced convulsions and died; only those with continuous SE for more than 4 h were eligible for the study (classification of seizures was based on Racine, 1972). Mice were placed in disposable cages in the same environment. Continuous behavioural observations were performed from SE cessation until the intended time points.

#### 2.1.3. Tissue Preparation for miRNA Isolation

All eligible mice were sacrificed by decapitation at different time points. Hippocampal brain tissue was collected and placed in an ice-cold phosphate-buffered saline (PBS) in a Petri dish. The DG was distinguishable from the CA by the gaps between them and was separated from the CA by inserting a sharp needle tip into each side of the DG. The needles were then slid superficially along the septotemporal axis of the hippocampus for isolation [18]. The isolated DG was picked up using a needle or forceps and placed in a sample tube containing RNAlater RNA stabilisation reagent (Qiagen, Dusseldorf, Germany). The CA and DG samples in all four groups were snap-frozen in liquid nitrogen and stored at −80 °C for further use.

### 2.2. Total RNA and miRNA Isolation

Each frozen sample was homogenised in 1 mL QIAzol lysis reagent (Qiagen, Dusseldorf, Germany). After the addition of 140 μL chloroform and centrifugation for 15 min, the upper phase was collected and precipitated with absolute ethanol (1.5 mL). The mixture was passed through an RNeasy Mini spin column and washed with RWT and RPE buffer, and total RNAs, including small RNAs, were eluted with RNase-free water. The quantity and quality of RNA were determined at 260/280 nm and 260/230 nm, respectively, using a NanoDrop spectrophotometer (Ocean Optics, Florida, USA) and an Agilent Bioanalyzer (Agilent, Santa Clara, CA, USA). The acceptable ratios for this study were a 260/280 ratio of >2.0 and a 230/280 ratio of >1.8. Acceptable RNA integrity numbers for this study were those greater than 7. The samples were chosen to meet these criteria for downstream assays.

### 2.3. SurePrint G3 Mouse miRNA Microarray

Microarray analysis was performed by a service provider (Genomax Technologies, Singapore) using the SurePrint G3 Mouse miRNA microarray kit (Agilent, Santa Clara, CA, USA). This kit was produced based on miRBase 16.0 and included 1023 mouse and 57 mouse viral G4859A miRNAs. The procedure included four steps: spiking in solution preparation, sample labelling and hybridisation, microarray washing, scanning, and feature extraction. With a sample input of 100 ng of total RNA, Agilent’s miRNA complete labelling and hybridisation kit generates fluorescently labelled miRNAs via ligation of one Cyanine 3-Cytidine biphosphate (pCp) molecule to the 3′ end of an RNA molecule with more than 90% efficiency. The miRNA microarray data were analysed using Genespring software (GX 11, Agilent, Santa Clara, CA, USA).

### 2.4. qRT-PCR Analysis of Chosen miRNAs

To validate the miRNA results, miRCURY LNATM universal cDNA synthesis and SYBR Green master mix kits (Exiqon, Copenhagen, Denmark) were used. First, total RNA, including small RNAs, was reverse-transcribed to cDNA. A real-time experiment was performed using 7900HT fast real-time PCR instruments (Life Technologies, Carlsbad, CA, USA). Cycle threshold (Ct) values were acquired using the software provided by the manufacturer. The reference gene of each group was RNU5G (the reference gene was chosen by pre-experimental tests; after some tests, this gene was expressed stably in each sample). The relative expression of each miRNA was calculated as ΔCt = Ct _target gene_ − Ct _internal control gene_. The relative quality (RQ) of mRNA from the SE group compared to that of the control group was calculated. Fold-changes in miRNA expression in the SE group were determined relative to the control using the 2^−∆∆Ct^ method. Statistical analyses were performed using one-way analysis of variance (ANOVA) with SPSS (version 25, IBM, Armonk, NY, USA).

### 2.5. Bioinformatics Analysis of the miRNA Microarray Results

Genes with significant differential expression between the three different stages of SE were first filtered using significance analysis of microarrays (SAM) software (*p* < 0.05). One-way ANOVA and the Benjamini-Hochberg test were used to perform multiple testing corrections.

Sequence checking of the gene expression dynamics clustering algorithm was used to analyse the gene expression time series and to determine the cluster set most likely to generate the observed time series. This method explicitly considers the dynamic nature of gene expression profiles during clustering and determines the number of different clusters.

According to the random variance model (RVM) corrective ANOVA, differentially expressed genes were selected in a logical sequence. A set of unique model expression tendencies was identified according to the different signal density change tendencies of genes under different situations. The raw expression values were converted into log2 ratios. Some unique profiles were defined using a strategy for clustering the short time-series gene expression data. The actual or expected number of genes assigned to each model profile was related to the expression model profiles. Analysed by Fisher’s exact test and multiple comparison test, the significant profiles exhibited higher probabilities than expected.

To further reduce the false positive rate (FDR), the intersection genes of TargetScan and miRanda were selected for target gene prediction. The intersecting target genes were annotated for gene function based on the gene ontology (GO) database for functional analysis. The GO terms which were structured and controlled were subdivided into three ontologies: molecular function (MF), biological process (BP), and cellular component (CC). The two-sided Fisher’s exact test and Benjamini-Honchberg test were used for multiple test correction, and FDR was used to adjust the *p*-values for multiple comparisons, with a *p* value < 0.01 and an adjusted *p* value < 0.01.

KEGG was used to detect disease-related pathways among the large list of miRNA-targeted genes. Pathway enrichment was calculated using hypergeometric distribution, and the *p*-values were adjusted for multiple comparisons by FDR.

## 3. Results

### 3.1. miRNA Microarray Analysis

Control versus days 3, 14, and 60 post-SE: in the CA area, 131 miRNAs were differently expressed, including 107 in the acute stage, 59 in the latent stage, and 48 in the chronic stage, 53, 49, and 26 miRNAs were upregulated and 54, 10, and 22 were downregulated, respectively. In the DG, 171 miRNAs were differentially expressed, including 114 in the acute stage, 90 in the latent stage, and 72 in the chronic stage, 36, 32, and 28 miRNAs were upregulated and 78, 58, and 44 were downregulated, respectively. Of these, 92 changed in both the CA and DG, 39 only in the CA, and 79 only in the DG area (cut off value: absolute value of fold change ≥ 1.5). The differentially expressed miRNAs at the three time points (days 3, 14, and 60) in the CA and DG are shown in Figure 1.

### 3.2. Validation of the miRNA Microarray Data by qRT-PCR

Primers of the selected miRNAs were used to validate the microarray data. The detailed results are listed in Table 1 and Table 2 and the miRNAs labelled with “*” were statistically different compared with those of control. Those labelled with “#” were statistically different compared with those of day 3 post-SE and those labelled with “^” were statistically different compared with those of day 14 post-SE. In the CA area, the validated miRNAs were 124, 137, 142-3p, 19a, 203, 27a, 494, 551b, 146a, 188-5p, and 193. In the DG area, the validated miRNAs were 124, 137, 142-3p, 19a, 203, 27a, 494, 551b, 132, 135b, 148a, 188-5p, and 672 (Table 1 and Table 2). Most qRT-PCR results were consistent with the microarray results, with the exception of those for miR-494 and 188-5p in both the CA and DG areas.

## 4. Bioinformatics Analysis of the miRNA Microarray Results

### 4.1. Gene Expression Trend Analysis

There were 26 miRNA gene expression trends on days 3, 14, and 60 post-SE in the CA and DG areas; they were randomly labelled with numbers. It can be concluded that trends of No.6 (A/D), 9 (B/E) and 19 (C/F) have significant differences in both CA (A, B, C) and DG (D, E, F) areas among the 26 trends generated (Figure 2). The miRNA expression levels were relatively similar between the two areas.

### 4.2. Gene Ontology Enrichment Analysis

The number of intersecting genes in the CA and DG predicted by Targetscan and miRanda is shown in Table 3.

GO enrichment analysis based on the predicted intersection target genes of differentially expressed miRNAs revealed that there are numerous functions regulated by differentially expressed miRNAs. These functions are regulated in both CA and DG areas (the larger the (*−LgP*) value, the smaller the *p* value, and the higher the significance level of GO).

Some common functions are regulated by miRNAs in the CA and DG. These are G protein-related processes (4), macromolecule-related processes (5), nitrogen compound-related processes (5), RNA-related processes (3), biosynthetic processes (3), metabolic processes (15), signalling (11), neuron differentiation (1), stimulus (11), transport processes (6), and development-related processes (37). The numbers in brackets represent the number of functions associated with the keywords. Table 4 lists the different functions regulated by the altered miRNAs in the CA and DG areas and their *−LgP* values.

### 4.3. Pathway Analysis

After deleting some obviously irrelevant pathways (those associated with cancer, infection, or other organs), the pathways regulated by the altered miRNAs in both the CA and DG areas were axon guidance; regulation of the actin cytoskeleton; focal adhesion; the Rap1, MAPK, Notch, Fc epsilon RI, TNF, and VEGF signalling pathways; dorso-ventral axis formation; glycerophospholipid metabolism; glycosaminoglycan biosynthesis-chondroitin sulphate/dermatan sulphate; the HIF-1 signalling pathway; endocrine and other factor-regulated calcium reabsorption; and the calcium signalling pathway. The different pathways regulated by the differentially expressed miRNAs in the CA and DG areas are listed in Table 5.

## 5. Discussion

miRNAs are small single-stranded RNAs approximately 22 nucleotides in length, which act as modulators of gene expression at the post-transcriptional level. It has been demonstrated that miRNA expression differs between brain regions [19,20], and it is believed that brain region-specific miRNA expression plays a role in regulating mRNA transcription. miRNAs have important effects on brain excitability and control several processes associated with epileptogenesis, including inflammation, synaptogenesis, and ion channel expression [8,21]. During the past decade, there have been some findings related to the specific miRNAs involved in epileptogenesis. However, seizure-evoked miRNA profiles differ depending on the species, mode of seizure induction, type of tissue analysed, and time of tissue collection after seizure [22]; hence, there might be various results among studies. We used the microarray method to screen for differentially expressed miRNAs in the pilocarpine-induced mouse epileptic model and qRT-PCR for validation. Control versus days 3, 14, and 60 post-SE: in the CA area, a total of 131 miRNAs were differentially expressed; 53, 49, and 26 miRNAs were upregulated, and 54, 10, and 22 were downregulated, respectively. In the DG area, a total of 171 miRNAs were differentially expressed; 36, 32, and 28 miRNAs were upregulated and 78, 58, and 44 were downregulated, respectively. Of these, 92 changed in both the CA and DG, 39 only in the CA, and 79 only in the DG area. The randomly selected miRNA qRT-PCR results were consistent with the microarray results, with the exception of those for miR-494 and 188-5p, which were excluded. We used bioinformatics to predict target genes, their functions, and related pathways. The differentially expressed miRNA target 11–1630 genes. Most of these proteins have multiple functions in epileptogenesis and are worthy of further study. There were 15 common pathways related to altered miRNAs; nine in the CA and seven in the DG area. To better understand the effect on epileptogenesis of these changed miRNAs, we compared our results with the previous literature; most were consistent with each other regardless of the models used, and some were consistent with the analysis of temporal lobe epilepsy (TLE) patient samples.

First, we compared our validated miRNA results with those of other studies. miR-27a-3p is overexpressed in the hippocampal cells of epileptic rats and KA-treated neurones [23]. Our results showed upregulation of miR-27a after SE in both the CA and DG areas at all three time points, consistent with a previous in vitro study. miR-132 was shown to be upregulated in a hippocampal neuronal culture and rat model of SE [24,25]. A study of the hippocampus of patients with TLE hippocampal sclerosis (TLE-HS) and the DG areas of electrodes in epileptic rats in the acute stage confirmed its upregulation [25]. Epileptic patients exhibit higher miR-132 expression in plasma, which could reflect the severity of epilepsy and predict the risks of complications [26]. miR-132 was upregulated in the DG area at the three time points after SE in this study. miR-203 was shown to be upregulated in the hippocampus of epileptic mice [27]. In addition, it was upregulated in the CA and DG areas at the three time points in this study. A previous study demonstrated upregulation of miR-146a expression in all hippocampal tissues in the latent and chronic stages of SE development in a rat model and in human TLE patients with hippocampal sclerosis [28]. Plasma miR-146a expression is increased in epileptic patients [26]. In our study, miR-146a was upregulated in the CA area at the three time points and in the DG area on days 3 and 14 post-SE. miR-142-3p is upregulated in the hippocampi of mTLE/HS patients of all age groups, including children, adolescents, and adults, and rats 3 months after SE [29]. In our study, miR-142-3p was upregulated in the CA area at the three time points and in the DG area on days 3 and 14 post-SE. Lower miR-137 expression has been associated with intellectual disability [30]. In our study, it was downregulated especially on day 3 post-SE in the CA region. Downregulated miR-551b-5p was found during long-term intermittent theta burst stimulation (iTBS) treatment of ischaemia/reperfusion (I/R), enhancing neurogenesis and migration [31]. In our study, miR-551b was downregulated on days 14 and 60 post-SE in the CA area and at the three time points in the DG area. miR-19a/b-3p, FoxO3, and SPHK1 have been shown to be upregulated, whereas SIRT1 has been shown to be downregulated in I/R [32]. We found that miR-19a was upregulated on day 3 post-SE compared to the control and downregulated on day 60 compared to on day 14 post-SE in both the CA and DG areas. miRNA-672 is downregulated and localised in neurones in the dorsal root ganglia (DRG) [33]. In our study, miR-672 was downregulated on days 14 and 60 post-SE in the DG area. Overexpression of miR-148a-3p in a febrile seizure (FS) model group affected the neuroimmune system and neuronal apoptosis in a previous study [34]. miR-148a-3p may serve as a novel diagnostic biomarker for TLE [35]. In our study, we found that miR-148a was downregulated in the DG region on days 14 and 60 post-SE. miR-135b-5p was shown to be downregulated in children with TLE and in an epileptic rat neuron model [36,37]. We found that miR-135b was upregulated on day 14 post-SE in the DG.

Second, some of our miRNA microarray results showed the same expression trends as in other studies, although they were not validated by qRT-PCR. Neuronal excitability [38]-related miRNAs include upregulated miR-23a [39] and downregulated miR-211 (DG) [40] and miR-128 (DG) [41]. Transcription factor-related miRNAs include downregulated miR-124 [42]. Apoptosis-related miRNAs include upregulated miR-21 [43], miR-34a (CA) [44], miR-129-5p [45], and downregulated miR-139-5p [46]. Ion channel-related miRNAs include downregulated miR-324-5p (DG) [47]. Oxidative stress-related miRNAs include upregulated miR-221 (DG) [48] and downregulated miR-153 [49], miR-221 (CA) [50], and miR-485 [51]. P2X7 related miRNAs include upregulated miR-22 [52], and some unclassed miRNAs such as downregulated miR-145-5p (DG) [53], miR-204 (DG), and miR-218 [17]. miRNAs without a label for CA or DG were miRNAs that changed in both regions. miRNAs changed only in the DG and had opposite expression trends compared to the whole hippocampus, which may play a protective role. For example, miR-592 was found to be upregulated in the DG area at days 14 and 60 post-SE in our study, suppressing its target p75NTR BDNF receptor, and inhibiting apoptotic signalling and neuronal death [54]. These miRNAs include miR-324-5p, 135b, 148a, and 221, etc. On the contrary, miRNAs changed only in the DG area and shared the same expression trend as those in the whole hippocampus, such as miR-128, 211, 145-5p, and 204, which may have proepileptic effects. For example, loss of miR-128 in dopamine 1 receptor (Drd1)-positive neurones leads to motor hyperactivity and lethal seizures [30].

Third, there are some miRNAs which have been reported to be related to SE, but for which we did not find statistically significant fold-changes, such as miR-134, 181a, 181a-5p, 181b, 155, 34c-5p, 935, 10a-5p, 142a-5p, 431-5p, 141, 184, 421, 25-3p, and 15a [38,46,48,55].

In the pathway-related analysis, miR-137 and 148-3p are related to the PI3K-Akt-mTOR pathway [34,56]. Mitogen-activated protein kinase 4 (MAP2K4) is a direct target of miR-27a-3p, and it is related to the MAPK signalling pathway [23]. miR-132 and miR-551b-5p are involved in the BDNF/TrkB pathway [24,31]. miR-203 directly suppresses the expression of MEF2C and promotes NF-kB, phosphorylated IkB/IKK, and inflammatory effectors through the MEF2C/NF-kB signalling pathway [57]. miR-19a/b-3p/SIRT1 promotes neuroinflammation by regulating FoxO3/SPHK1 signalling, leading to cell death [32]. miR-146a and 142-3p are involved in the TLR signalling pathway [58,59,60]. miR-672 and 135b-5p are related to the pyroptosis pathway [36,37,61]. Among those, the PI3K-Akt-mTOR and MAPK signalling pathways were validated and changed the miRNAs-related pathway, which was consistent with our bioinformatics analysis.

## 6. Conclusions

We studied stage and subfield-associated hippocampal miRNA expression patterns after pilocarpine-induced SE and found differences in the CA and DG areas. This is important for understanding the pathogenesis of this disease in these two areas, as the loss of neurones in the DG is less severe than that in the CA. The number of animals should be increased in in-depth studies. Some of our results were consistent with other studies, regardless of the model, and these miRNAs might be more valuable for further studies, specifically miR-146a, miR-132, and miR-137. All of them are candidate therapeutic targets.

## Figures and Tables

**Figure 1 biomedicines-10-03012-f001:**
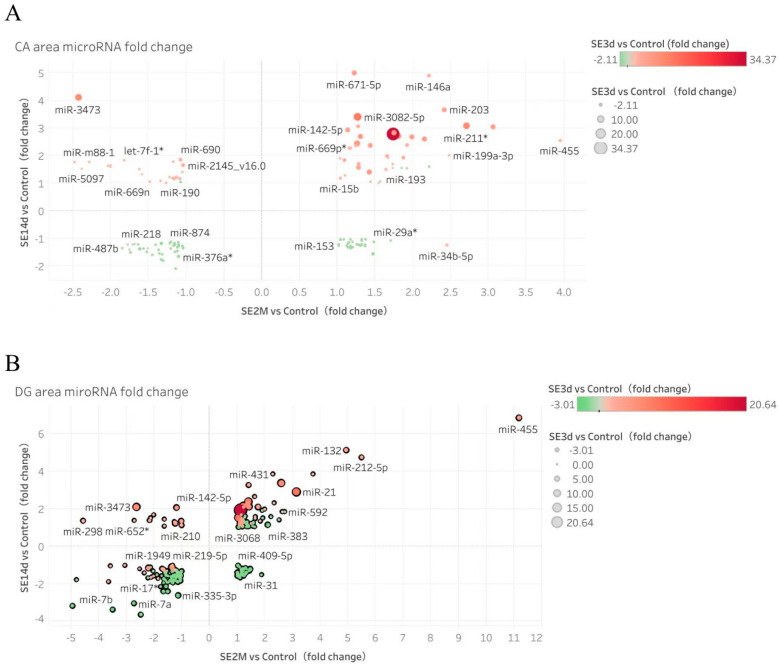
Differently expressed miRNAs in the CA (**A**) and DG (**B**) areas over three time points. The fold-change value on day 14 post-SE was used as the ordinate and the fold-change value on day 60 post-SE as the abscissa. The colour gradient (from green to red) and the size of dots were used to represent the fold-change value on day 3 post-SE. miR-494 and 188-5p were excluded according to the validation results. Tableau software was used to map the changes in the expression of different miRNAs. Note: the * in the figure like miR-29a* has no other meaning but the name of a specific miRNA.

**Figure 2 biomedicines-10-03012-f002:**
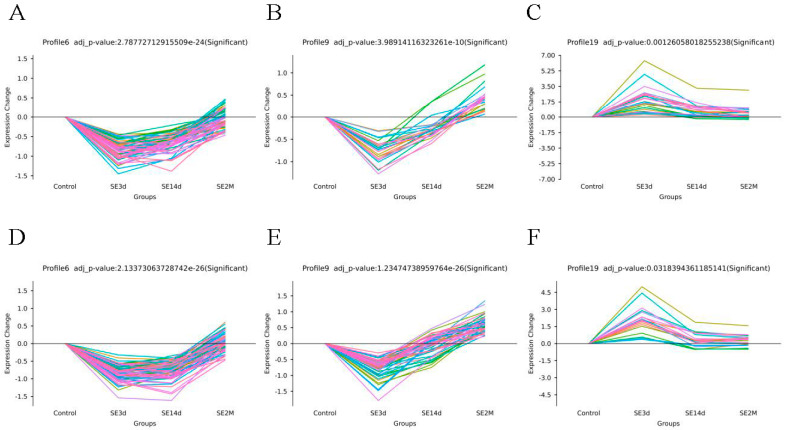
CA (**A**–**C**) and DG (**D**–**F**) area of significant gene expression trends. The abscissa represents different time points, the ordinate represents the grading of gene changes, and each line represents a different miRNA. It can be concluded that trends of No.6 (A/D), 9 (B/E) and 19 (C/F) have significant differences in both CA and DG areas among the 26 trends generated.

**Table 1 biomedicines-10-03012-t001:** Validation of microarray results in the CA area by qRT-PCR.

Primer Name	Group	qRT-PCR Fold-Change	Microarray Fold-Change	*F*	*P*
miR-124	Control	1 ± 0.25	1	1.693	0.245
	Day 3	0.64 ± 0.2	−1.745579		
	Day 14	0.89 ± 0.3	−1.1375024		
	Day 60	0.68 ± 0.12	−1.1038424		
miR-137	Control	1 ± 0.15	1	2.154	0.172
	Day 3	0.62 ± 0.23 *	−2.0571318		
	Day 14	0.78 ± 0.1	−1.3803179		
	Day 60	0.75 ± 0.23	−1.619207		
miR-142-3p	Control	1 ± 0.11	1	14.289	0.001
	Day 3	5.32 ± 0.81 *	5.619121		
	Day 14	3.42 ± 1.64 *#	2.8325222		
	Day 60	1.34 ± 0.19 #	1.7457325		
miR-19a	Control	1 ± 0.34	1	25.305	0.000
	Day 3	2.87 ± 0.19 *	2.878858		
	Day 14	1.17 ± 0.52 #	1.1613489		
	Day 60	0.71 ± 0.17 #	−1.1601521		
miR-203	Control	1 ± 0.36	1	8.605	0.007
	Day 3	4.5 ± 1.14 *	4.5549407		
	Day 14	3.44 ± 0.77 *	3.667932		
	Day 60	1.89 ± 1.18 #	2.421915		
miR-27a	Control	1 ± 0.16	1	2.477	0.136
	Day 3	1.39 ± 0.14	1.7447702		
	Day 14	1.57 ± 0.41 *	1.6940101		
	Day 60	1.32 ± 0.25	1.7312951		
miR-494	Control	1 ± 0.62	1	0.272	0.844
	Day 3	1.02 ± 0.35	7.6855636		
	Day 14	1.05 ± 0.58	2.0513108		
	Day 60	0.73 ± 0.32	−1.0420982		
miR-551b	Control	1 ± 0.21	1	2.848	0.105
	Day 3	0.59 ± 0.28	−1.9337255		
	Day 14	0.53 ± 0.14 *	−1.7970246		
	Day 60	0.54 ± 0.27 *	−1.3070583		
miR-146a	Control	1 ± 0.34	1	2.660	0.119
	Day 3	2.99 ± 0.86	3.089705		
	Day 14	4.86 ± 3.23 *	4.8941493		
	Day 60	1.69 ± 1.31	2.211484		
miR-188-5p	Control	1 ± 0.1	1	6.129	0.018
	Day 3	1.23 ± 0.18	333.90417		
	Day 14	0.89 ± 0.22 #	22.7263		
	Day 60	0.69 ± 0.09 *#	26.388815		
miR-193	Control	1 ± 0.26	1	0.073	0.973
	Day 3	1.02 ± 0.38	2.0996573		
	Day 14	1.13 ± 0.44	1.5109724		
	Day 60	1.09 ± 0.43	1.6478502		

Note: * indicates comparison to control, *p* < 0.05; # indicates comparison to day 3 post-SE, *p* < 0.05.

**Table 2 biomedicines-10-03012-t002:** Validation of microarray results in the DG area by qRT-PCR.

Primer Name	Group	qRT-PCR Fold-Change	Microarray Fold-Change	*F*	*P*
miR-124	Control	1 ± 0.12	1	2.225	0.163
	Day 3	0.55 ± 0.2	−1.5798804		
	Day 14	0.81 ± 0.16	−1.3798828		
	Day 60	0.58 ± 0.4	−1.1080296		
miR-137	Control	1 ± 0.25	1	0.604	0.630
	Day 3	0.77 ± 0.08	−1.832393		
	Day 14	0.88 ± 0.22	−1.588261		
	Day 60	1.03 ± 0.41	−1.2935005		
miR-142-3p	Control	1 ± 0.07	1	19.760	0.000
	Day 3	6.4 ± 1.82 *	4.912		
	Day 14	2.75 ± 0.9 #	2.225		
	Day 60	0.72 ± 0.21 #^,^^	1.392988		
miR-19a	Control	1 ± 0.35	1	5.542	0.024
	Day 3	2.61 ± 1.08 *	2.1299305		
	Day 14	1.23 ± 0.48 #	−1.2170526		
	Day 60	0.54 ± 0.43 #	−1.7115029		
miR-203	Control	1 ± 0.67	1	2.745	0.113
	Day 3	7.57 ± 5.08 *	5.519774		
	Day 14	3.82 ± 1.32	3.3713422		
	Day 60	2.43 ± 2.6	2.5886562		
miR-27a	Control	1 ± 0.31	1	2.385	0.145
	Day 3	1.6 ± 0.28	1.5262655		
	Day 14	1.61 ± 0.48	1.5529257		
	Day 60	0.96 ± 0.5	1.6639766		
miR-494	Control	1 ± 0.28	1	0.764	0.545
	Day 3	1.28 ± 0.43	4.505004		
	Day 14	1.51 ± 0.43	1.3652874		
	Day 60	1.17 ± 0.52	−1.180082		
miR-551b	Control	1 ± 0.1	1	12.988	0.002
	Day 3	0.63 ± 0.23 *	−1.7511983		
	Day 14	0.32 ± 0.04 *^,^#	−3.644983		
	Day 60	0.32 ± 0.19 *^,^#	−2.47584		
miR-132	Control	1 ± 0.23	1		
	Day 3	3.38 ± 0.81	2.6384544	5.801	0.021
	Day 14	6.55 ± 1.34 *	5.115643		
	Day 60	6.28 ± 3.43 *	4.9513855		
miR-135b	Control	1 ± 0.35	1	3.726	0.061
	Day 3	1.56 ± 0.32	1.2437161		
	Day 14	4.86 ± 2.21 *^,^#	3.8363545		
	Day 60	3.77 ± 2.37	3.7484105		
miR-148a	Control	1 ± 0.17	1	9.789	0.005
	Day 3	0.83 ± 0.36	−1.731051		
	Day 14	0.34 ± 0.17 *^,^#	−3.1871736		
	Day 60	0.17 ± 0.09 *^,^#	−4.944259		
miR-188-5p	Control	1 ± 0.16	1	1.971	0.197
	Day 3	1.27 ± 0.05	28.367397		
	Day 14	0.79 ± 0.27	3.5954535		
	Day 60	0.63 ± 0.6	2.1866686		
miR-672	Control	1 ± 0.07	1	3.806	0.058
	Day 3	0.81 ± 0.26	−1.6016815		
	Day 14	0.59 ± 0.11 *	−1.9541458		
	Day 60	0.52 ± 0.2 6 *	−1.6448656		

Note: * indicates comparison to control, *p* < 0.05; # indicates comparison to day 3 post-SE, *p* < 0.05; ^ indicates comparison to day 14 post-SE, *p* < 0.05.

**Table 3 biomedicines-10-03012-t003:** Numbers of predicted intersecting target genes of changed miRNAs.

Changed miRNA in Both the CA and DG Areas	Predicted Intersecting Target Gene Number	Changed miRNA in DG Area	Predicted Intersecting Target Gene Number	Changed miRNA in CA Area	Predicted Intersecting Target Gene Number
miR-124	1630	miR-128	828	miR-193	164
miR-127	20	miR-132	305	miR-487b	11
miR-129-5p	391	miR-133b	476	miR-93	932
miR-136	170	miR-143	273		
miR-137	912	miR-145	552		
miR-139-5p	289	miR-195	940		
miR-142-3p	252	miR-204	396		
miR-149	288	miR-211	396		
miR-150	201	miR-28	192		
miR-153	599	miR-31	271		
miR-186	504	miR-33	313		
miR-190	119	miR-346	84		
miR-203	608	miR-381	548		
miR-20b	935	miR-383	115		
miR-21	230	miR-384-3p	152		
miR-210	27	miR-431	87		
miR-218	757	miR-488	210		
miR-22	370	miR-497	940		
miR-221	308	miR-873	178		
miR-222	308				
miR-223	220				
miR-25	725				
miR-324-5p	85				
miR-326	268				
miR-328	123				
miR-335-5p	175				
miR-362-3p	225				
miR-370	245				
miR-377	364				
miR-379	67				
miR-384-5p	1078				
miR-410	434				
miR-411	121				
miR-433	196				
miR-485	489				
miR-494	314				
miR-496	77				
miR-874	194				
miR-9	1017				

**Table 4 biomedicines-10-03012-t004:** Different functions regulated by changed miRNAs in CA and DG areas.

GO Name (CA Area)	−*LgP*	GO Name (DG Area)	−*LgP*
myeloid cell differentiation	4.79	sensory perception of temperature stimulus	5.17
positive regulation of cell differentiation	4.24	head development	4.72
macromolecule de-acylation	4.10	response to metalation	4.47
amyloid precursor protein metabolic process	4.06	modification-dependent macromolecule catabolic process	4.26
positive regulation of multi-organism process	3.87	RNA localisation	3.93
prostanoid metabolic process	4.03	stem cell division	4.09
membrane biogenesis	4.06	ossification	4.07
cellular protein complex assembly	3.82	adrenergic receptor signalling pathway	3.88
ganglion development	3.69	cell aggregation	3.74
membrane docking	3.55	detection of biotic stimulus	3.72
response to peptide	3.55	developmental growth involved in morphogenesis	3.70
suckling behaviour	3.82	cell-cell adhesion via plasma-membrane adhesion molecules	3.58

**Table 5 biomedicines-10-03012-t005:** Different pathways regulated by the changed miRNAs in CA and DG areas.

CA Area-Specific Pathways	−*LgP*	DG Area-Specific Pathways	−*LgP*
Ras signalling pathway	10	Hippo signalling pathway	10
Hedgehog signalling pathway	4.60	PI3K-Akt signalling pathway	10
Glycosaminoglycan biosynthesis—heparan sulphate/heparin	3.95	Tight junction	4.22
Synaptic vesicle cycle	3.77	Cholinergic synapse	3.98
Adipocytokine signalling pathway	3.61	N-Glycan biosynthesis	3.27
Chemokine signalling pathway	3.32	Glycosaminoglycan biosynthesis—keratan sulphate	2.51
B cell receptor signalling pathway	3.14	Renin secretion	2.44
ECM-receptor interaction	2.81		
Jak-STAT signalling pathway	2.43		

## Data Availability

The data presented in this study are available on request from the corresponding author.

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
