# Peer review of "Stage- and Subfield-Associated Hippocampal miRNA Expression Patterns after Pilocarpine-Induced Status Epilepticus"

_biomedicines, 2022, doi:10.3390/biomedicines10123012_

Round 1

Reviewer 1 Report

Thank you for asking me to review this manuscript. The authors have used evaluation of miRNAs to characterize pathophysiological processes associated with pilocarpine induced hippocampal injury in mice. This manuscript represents a lot of work that has been rigorously carried out. I do not have serious methodological concerns although I am not an expert in the use of these tools. However, there are conceptual and interpretation issues that deserve further consideration.

1.       It is not clear whether any of the mice developed epilepsy. It would be unusual for spontaneous seizures to occur by D3 post pilocarpine, but some are likely to have developed seizures by D14 and most by D60. However, this is not universally true. The authors should discuss whether the presence or absence of epilepsy could impact their findings. The conclusion in the abstract makes a statement about epileptogenesis that is difficult to sustain in the absence of knowledge about seizures. This is particularly important in mouse as the mouse model is less robust in terms of outcomes compared to rat models

2.       There are many changes identified with clear trajectories over time. There are also many biological processes identified with the GO and KEGG approaches. It is not clear how those processes interact with each other or result in a disease phenotype. The discussion identifies papers that have miRNA findings in common with the current paper and that is useful. However, it is not possible to construct a model of the biological changes that occur in a more holistic way. Some changes will be a direct result of the intervention, some a result of brain injury and neuronal death, some will be a function of neuronal reorganization etc. I think a ‘deeper dive’ into network (interaction) levels will be important.

3.       Without clearer direction on which processes identified are novel and biologically plausible it is difficult to see how these data can be used to drive future research endeavors.

Author Response

Response letter

1.It is not clear whether any of the mice developed epilepsy. It would be unusual for spontaneous seizures to occur by D3 post pilocarpine, but some are likely to have developed seizures by D14 and most by D60. However, this is not universally true. The authors should discuss whether the presence or absence of epilepsy could impact their findings. The conclusion in the abstract makes a statement about epileptogenesis that is difficult to sustain in the absence of knowledge about seizures. This is particularly important in mouse as the mouse model is less robust in terms of outcomes compared to rat models.

Thanks for your question. In order to understand the significance of this work, it is necessary to understand the model itself. The following oppion is obtained from a review “The pilocarpine model of temporal lobe epilepsy” (Curia et al., 2008), And I have already quote in the article for the readers to better understand the work.

Understanding the pathophysiogenesis of temporal lobe epilepsy (TLE) largely rests on the use of models of status epilepticus (SE), as in the case of the pilocarpine model. The main features of TLE are: (i) epileptic foci in the limbic system; (ii) an “initial precipitating injury”; (iii) the so-called “latent period”; and (iv) the presence of hippocampal sclerosis leading to reorganization of neuronal networks.

Injection of pilocarpine induces a SE that is characterized by tonic-clonic generalized seizures. After several hours of SE, pilocarpine-treated animals remit spontaneously and go into a seizure-free period, known as latent period, before displaying the SRSs that characterize the chronic epileptic condition.

Following SE induced by pilocarpine (340 mg/kg), adult (25–30 g) male albino mice, mortality rate was 25–50%, mean latent period was 14.4 days, spontaneous recurrent seizures (SRSs) lasted 50–60 s and they occurred with a frequency of 1-5 seizures per week (Cavalheiro et al., 1991 vs. 1996 and Turski et al., 1984).

2.There are many changes identified with clear trajectories over time. There are also many biological processes identified with the GO and KEGG approaches. It is not clear how those processes interact with each other or result in a disease phenotype. The discussion identifies papers that have miRNA findings in common with the current paper and that is useful. However, it is not possible to construct a model of the biological changes that occur in a more holistic way. Some changes will be a direct result of the intervention, some a result of brain injury and neuronal death, some will be a function of neuronal reorganization etc. I think a ‘deeper dive’ into network (interaction) levels will be important.

This is also a very good question. We have done this kind of analysis as follows:

We use the relationship between miRNA and its target Gene establishing miRNA-Gene action network. In miRNAs and Gene networks Genes were marked by circle, miRNAs by rounded rectangle. The center of the network was with high network center degree. The highest Degree of the miRNAs or Genes in the core position in the network will be marked.

miRNA Target Go Network was built according to the relationships of significant GOs and genes, as well as the relationships among miRNAs and GOs. In the miRNA Target Go Network, a circle represents a GO, a square represents a miRNA, and relationships between them are represented by edges. Using the methods of graph theory, we evaluated the regulatory status of miRNAs and GOs; the evaluation criteria were the degrees of miRNAs and GOs in the network. The degree of each miRNA was the number of GOs regulated by that miRNA, and the degree of each GO was the number of miRNAs which regulated the GO. Key miRNAs and GOs in the network had the highest degrees.

But the network is very complex, so far, we can not explain it properly and due to limited space we did not put this part of data in this article.

3.Without clearer direction on which processes identified are novel and biologically plausible it is difficult to see how these data can be used to drive future research endeavors.

â‘ This is a preliminary screening of the intrested miRNA, each miRNA may have multifunctions as they can target 11 to 1630 genes. According to the literature review, they are involved in neuroinflammation, pathological circuit reformation, apoptosis, dysfunction of glia cells, oxidative stress, autophagy, deregulation of neurotrophic factors, blood brain barrier (BBB) dysfunctions, apoptosis, ion channel dysregulation, axonal guidance, and synaptic plasticity etc (Wang et al., 2021; Zummo et al., 2021). Each of them worth further studies. They are pathogenetic involved and can be diagnostic biomarkers and finally used as therapeutic agent development.

â‘¡As previous studies have mostly been performed only in a single model or species, the results may need to be verifified in models representing difffferent etiologies or in larger animals. Our study represents the result of one model, all those differentially expressed miRNAs may be a potential target for further studies.

And overall, I made the necessary revision in the article including language editting, please refer to the articles for the details.

Reviewer 2 Report

Review of “Stage- and subfield-associated hippocampal miRNA expression 2 patterns in pilocarpine-induced status epilepticus”

I found this manuscript easy to read and well explained. The flow in presentation tells a nice story to follow. It is very interesting to note the miRNA expression patterns in various hippocampal regions (CA & DG).  Examining the profiles at various times after the induction of status epilepticus adds a nice touch to know how long some of the changes still occur.

I do  not see any major issues which need to be addressed.  I only have minor suggestions.

Minor:

1.       Line 52… pilocarpine.  Where the drug is introduced to the reader it might be useful to a reader how pilocarpine induces status epilepticus at a mechanistic level (i.e., as a muscarinic receptor agonist, etc…). Of course, people in this research area understand but just a short explanation of the mechanism helps to think about how this can induce neural activity.

So, if one was comparing cells in culture to this research one might think just depolarizing cells with KCl would be equivalent to pilocarpine.

2.       Figure 2 . Up to the authors  but think 2 columns instead of 2 rows would allow the graphs to be enlarged and easy to examine. But I understand the tables show enough detail.

Author Response

1.line 52… pilocarpine.Where the drug is introduced to the reader it might be useful to a reader how pilocarpine induces status epilepticus at a mechanistic level (i.e., as a muscarinic receptor agonist, etc…). Of course, people in this research area understand but just a short explanation of the mechanism helps to think about how this can induce neural activity.

So, if one was comparing cells in culture to this research one might think just depolarizing cells with KCl would be equivalent to pilocarpine.

Thanks for your advice. This is a good question for me to ensure the readers can understand the mechanism of this model and this part had been revised in the article referring to the following sentence:

The ability of pilocarpine to induce SE is likely to depend on activation of the M1 muscarinic receptor subtype, since M1 receptor knockout mice do not develop seizures in response to pilocarpine (Hamilton et al., 1997).

2.Figure 2 . Up to the authorsbut think 2 columns instead of 2 rows would allow the graphs to be enlarged and easy to examine. But I understand the tables show enough detail.

Revised in the article. Made necessary changes to the graph to make sure labels in the figure 2 can be visible.

Thanks a lot.

Reviewer 3 Report

The paper can contain interesting data but there is some point to be addressed for understanding if the findings could be interesting:

-based on the data enclosed in the article the journal namely biomedicines seems to be not appropriate since in the paper there is no mention of drugs therapeutics or biotherapeuthics. Another journal such as Biology of mdpi in my opinion should be more appropriate.

-number of approval for the in vivo study should be provided in materials and methods

-number of animals should be reported in materials and methods section. Through the manuscript I have no read the number of animals used and if the number is sufficient for a serious and rigorous statistical analysis

-Figure 1 should be enlarged or the label should be enlarged for allowing the read of the audience

-labels in the figure 2 should be visible

-Discussion first row. authors wrote miRNA is small molecule. This sentence is wrong miRNA is not a small molecule. please correct and correct through the text if necessary

-Authors should add a conclusion section for highlighting and summarizing the work and the main findings of the paper. Furthermore the limitation of the study and how this results should be useful for other scientists should be described

Author Response

The paper can contain interesting data but there is some point to be addressed for understanding if the findings could be interesting:

-based on the data enclosed in the article the journal namely biomedicines seems to be not appropriate since in the paper there is no mention of drugs therapeutics or biotherapeuthics. Another journal such as Biology of mdpi in my opinion should be more appropriate.

Thanks for your suggestions. I have revised in the article and emphasized the contribution of this study for the  therapeutic targets screening of epilepsy. And English language editting has been done.

-number of approval for the in vivo study should be provided in materials and methods

IACUC approved protocol No:B27/09 &098/09

Revised in the article.

-number of animals should be reported in materials and methods section. Through the manuscript I have no read the number of animals used and if the number is sufficient for a serious and rigorous statistical analysis

As stated in the article there is 3 animals in each group.

-Figure 1 should be enlarged or the label should be enlarged for allowing the read of the audience

-labels in the figure 2 should be visible

Revised in the article.

-Discussion first row. authors wrote miRNA is small molecule. This sentence is wrong miRNA is not a small molecule. please correct and correct through the text if necessary

Revised in the article.

-Authors should add a conclusion section for highlighting and summarizing the work and the main findings of the paper. Furthermore the limitation of the study and how this results should be useful for other scientists should be described

Revised in the article.

Round 2

Reviewer 3 Report

authors addressed my concerns providing a revised version suitable for the publication